# Microarray Analysis Reveals Changes in tRNA-Derived Small RNAs (tsRNAs) Expression in Mice with Septic Cardiomyopathy

**DOI:** 10.3390/genes13122258

**Published:** 2022-11-30

**Authors:** Ludong Yuan, Yuting Tang, Leijing Yin, Xiaofang Lin, Zhengyang Luo, Shuxin Wang, Jing Li, Pengfei Liang, Bimei Jiang

**Affiliations:** 1Department of Pathophysiology, Sepsis Translational Medicine Key Laboratory of Hunan Province, National Medicine Functional Experimental Teaching Center, Xiangya School of Medicine, Central South University, Changsha 410008, China; 2Department of Burns and Plastic Surgery, Xiangya Hospital, Central South University, Changsha 410008, China

**Keywords:** sepsis, septic cardiomyopathy, tRNA-derived small RNAs (tsRNAs), angiogenin (ANG), microarray analysis

## Abstract

**Background:** tRNA-derived small RNAs (tsRNAs) as a novel non-coding RNA have been studied in many cardiovascular diseases, but the relationship between tsRNAs and septic cardiomyopathy has not been investigated. We sought to analyze changes of the expression profile of tsRNAs in septic cardiomyopathy and reveal an important role for tsRNAs. **Methods:** We constructed a sepsis model by cecal ligation and puncture (CLP) in mice, and microarray analysis was used to find differentially expressed tsRNAs. Quantitative real-time PCR was used to verify the expression of tsRNAs and the interference effect of angiogenin (ANG), a key nuclease producing tsRNAs. Bioinformatics analysis was used to predict target genes and functions. CCK-8 and LDH release assays were used to detect cell viability and cell death. **Results:** A total of 158 tsRNAs were screened, of which 101 were up-regulated and 57 were down-regulated. A total of 8 tsRNAs were verified by qPCR, which was consistent with microarray results. Gene Ontology (GO) annotation and Kyoto Encyclopedia of Genes and Genomes (KEGG) pathway enrichment analyses suggest that these tsRNAs may be associated with the Wnt signaling pathway and participate in cellular process. The expression of tsRNAs decreased after the interference of the key nuclease ANG, while CCK-8 suggested a corresponding decrease in cell viability and an increase in the release of LDH (cell death), indicating that tsRNAs can protect cardiomyocytes during the development of septic cardiomyopathy, reduced cardiomyocyte death. **Conclusions**: A total of 158 tsRNAs changed significantly in septic cardiomyopathy, and these tsRNAs may play a protective role in the development of septic cardiomyopathy.

## 1. Introduction

Sepsis is defined as a life-threatening organ dysfunction that results from a host’s maladjusted response to infection [1]. Sepsis-induced cardiomyopathy (SICM) or sepsis-induced myocardial dysfunction (SIMD) are common manifestations of transient heart failure in patients with sepsis [2]. Over the past 50 years, numerous studies have shown that myocardial dysfunction is common in patients with sepsis, with approximately 50% of patients with sepsis exhibiting signs of myocardial dysfunction [3]. However, once septic cardiomyopathy is developed, the cardiovascular failure caused by septic cardiomyopathy will decrease the whole blood circulation and aggravate tissue hypoxia, mitochondrial dysfunction, and tissue metabolic dysfunction; eventually, the patient will die of septic shock [4]. The latest epidemiological data show that sepsis-related deaths account for 19.7% of all deaths worldwide [5], and sepsis costs more than $24 billion annually in the United States [6]. Although sepsis incidence and mortality have been declining since 1990, it remains one of the leading causes of health loss worldwide. The characteristics of high mortality and poor prognosis have made sepsis and septic cardiomyopathy a concern of researchers since the 1980s when they were described. The study of its pathogenesis provides a new way for us to understand the disease systematically. In recent years, with the rise of research on small non-coding RNA (snoRNA), we begin to think about the relationship between the snoRNA and septic cardiomyopathy.

Since Hoagland, Zamecnik, and Stephenson (1957) discovered the amino-acid-transfer ribonucleic acid (t-RNA) or soluble RNA (s-RNA), we have been working to demonstrate that tRNAs play a central role in the translational mechanism of protein synthesis [7]. It was not until the beginning of this century that we noticed that tRNA could perform other biological functions by producing tRNA-derived small RNA (tsRNA) [8]. tsRNA is a new class of 18–40 nt in length [9]. Studies have shown that they are generated not by random degradation of tRNA but by cleavage by specific nucleases, mainly including Elac Ribonuclease Z 2 (ELAC2)/RNase Z, RNase L, dicer, and angiogenin (ANG). Dicer, as an RNase III-type protein, is involved in the production of microRNA and siRNA [10], but the interesting thing is that ANG is also involved in the generation of tsRNA. Since ANG was first described by Professor Vallee of Harvard University in 1985 [11], we have also paid more attention to its role in angiogenesis as the first human tumor-derived protein found to stimulate vascular growth [12]. It was not until 2009 that researchers were concerned that, in addition to the function of inducing angiogenesis, ANG could be involved in tRNA cleavage under stress conditions, generating tsRNAs that in turn promote translational inhibition [13,14]. Under the action of nucleases, we divided tsRNAs into two large classes, tRNA-derived fragments (tRFs) and tRNA-derived stress-induced RNAs (tiRNAs), according to the position of the tsRNAs derived from the parental tsRNA-precursors. More specifically, tRFs can be further classified into several types: tRF-1, tRF-2, tRF-3, tRF-5 and i-tRF; and tiRNAs can be classified into 5′ tiRNAs and 3′ tiRNAs. [15]. ANG is mainly involved in the generation of tiRNAs [16]. In addition, the functions of tsRNA are diverse, including regulation of gene expression regulation, participation in epigenetic regulation, promote virus replication, initial virus reverse transcription, anti-apoptosis, translation inhibition, and cell-to-cell communication [17]. Although the function of tsRNA has been explored in various fields, it is still mainly focused on tumors, and the related function of tsRNA has not been explored in the development of septic cardiomyopathy. Therefore, we used microarray hybridization to explore the potential link between tsRNA and septic cardiomyopathy. The microarray results indicated that 158 tsRNAs were significantly changed in the development of septic cardiomyopathy, 101 of which were up-regulated and 57 which were down-regulated. Further bioinformatics analysis suggested that the target genes of these tsRNAs included Yipf6 and NCOA4, and the main enriched signaling pathways included the MAPK pathway and the Wnt pathway. Furthermore, through CCK-8 and LDH release assays, we preliminarily observed that these tsRNAs can enhance cardiomyocyte activity and reduce cardiomyocyte death. We hypothesized that these tsRNAs may have a protective effect on cardiomyocytes during the development of septic cardiomyopathy.

## 2. Materials & Methods

### 2.1. Animals and Experimental Groups

C57BL/6J male wild type (WT) mice (each 8–10 weeks old and weighing 20–25 g) were purchased from the Department of Central South University Laboratory Zoology (Changsha, China), and all animal experiments were approved by the IRB of Third Xiangya Hospital, Central South University, and the approval number is 2019-s218. The mice were reared with strict specific-pathogen-free (SPF) conditions (room temperature at 25 ℃ and 12 h light–dark cycle), and had free access to standard rodent water and food. After one week of acclimatization, the mice were randomly separated into three groups: SHAM, CLP 12 h, and CLP 24 h groups. A total of 42 mice were used in this study, 12 in the SHAM group (*n* = 3 for microarray analysis, *n* = 6 for qPCR and myocardial enzymatic assays, *n* = 3 for tissue sectioning), 12 in the CLP 12 h group (*n* = 8 for qPCR and myocardial enzymatic assays, *n* = 3 for tissue sectioning, 1 mouse died), and 18 in the CLP 24 h group. (*n* = 3 for microarray analysis, *n* = 8 for qPCR and myocardial enzymatic assays, *n* = 3 for tissue sectioning, 4 mice died). The number of mice in each group was determined based on published studies [18] and the needs of our own experimental design. We followed 4–5 mice/cage and distinguished the three different groups of mice by the different positions and the hanging tags indicated on the cages, thus avoiding confusion. After they were anesthetized by isoflurane, cecal ligation and puncture (CLP) was used to establish the model of septic cardiomyopathy. In the model, we used a 5 mL syringe needle for cecal ligation and wound suture with No. 4 surgical suture, 0.9% NaCl (50mL/kg) was injected subcutaneously into the back skin immediately after modeling, and the modeling times were 12 and 24 h. The SHAM Operation Group was consistent with the CLP group except that the cecum was not ligated and punctured. At the end of the modelling, we observed that the mice in the experimental group had pus in the corner of the eye, decreased body temperature, depression, and reduced mobility, which met the criteria for euthanasia, so we used isoflurane to deeply anaesthetize all the mice that met the experimental criteria and collected hearts and serum after cervical dislocation.

### 2.2. Cell Culture

H9C2 cells were obtained from the Shanghai Cell Bank of the Chinese Academy of Sciences (Shanghai, China) and maintained in Dulbecco’s modified Eagle medium supplemented with 10% fetal bovine serum. The cells were cultured at 37 °C in a humidified incubator with 5% CO_2_.

### 2.3. Cell Transfection Experiments

For small interfering RNA (siRNA) transfection, ANG-specific siRNA (si-ANG), and the corresponding negative control siRNA (siCon) (RiboBio, GuangZhou, China) were transfected into cells. Reagent (RiboBio) was used for cell transfection, according to the manufacturer’s instructions. After 48 h of transfection, the H9C2 cells were lysed for further experiments.

### 2.4. Cell Viability

Cell viability was assessed using the Cell Counting Kit-8 assay (CCK-8; Beyotime institute of Biotechnology, Shanghai, China). Cells were seeded at a density of 2–4 × 103 cells/well into 96-well plates. The medium was replaced with 100 μL of complete culture medium with or without LPS (10 ug/mL) and TNF-α (25 ng/mL) after 48 h of transfection. Processing 24 h, the CCK-8 reagent (1:100) was added into the wells, followed by incubation for 1 h. The absorbance of the samples at 450 nm was measured using a microtiter reader. All experiments were performed in triplicate.

### 2.5. Lactate Dehydrogenase Assay

After cell death, lactate dehydrogenase (LDH) was released into the supernatant, we used the LDH cytotoxicity detection kit (Beyotime Institute of Biotechnology, China) to monitor LDH released by H9C2 cells into the medium to determine cytotoxicity. LDH release reagent treatment (1:10 dilution, 1 h) was used as a positive control to test maximum LDH release. The optical density was measured spectrophotometrically at 490 nm on a microplate reader [19].

### 2.6. Array Hybridization

Briefly, 100 ng of total RNA extracted in each sample in SHAM versus CLP groups was first dephosphorylated to form 3-OH ends. DMSO was added and heated to 100 °C for 3 min to denature the RNAs and chilled immediately to 0 °C. RNA end labeling was performed by adding ligase buffer, BSA, final 50 mM pCp-Cy3, and 15 units T4 RNA ligase in a 28 uL reaction at 16 °C overnight. We then hybridized on a mouse tsRNA microarray (8 × 15K, Arraystar) containing a 1767 tsRNA probe. The slides were scanned on an Agilent G2505C microarray scanner.

### 2.7. Data Analysis

The obtained microarray results were analyzed by Agilent Feature Extraction software (version 11.0.1.1). First of all, the raw intensities were log2 transformed and quantile normalized. After normalization, the probe signals having Present (P) or Marginal (M) QC flags in a certain proportion were retained. Then, multiple probes from the same tsRNA were averaged and consolidated into one RNA level. The difference in tsRNA expression between the two groups was determined by fold change (FC) and statistical significance (*p* value) thresholds. *p* values < 0.05 and FC ≥ 1.5 were used as intervals for the comparison of differences in tsRNA between the two groups. Finally, hierarchical clustering was performed and volcano plots and scatter plots were drawn to show the distinguishable tsRNA expression patterns among samples.

### 2.8. RNA Extraction

RNA was extracted from myocardial tissue in both SHAM groups as well as in CLP, and total RNA in myocardial tissue was extracted using TRIzol (Invitrogen, Carlsbad, CA, USA) as required by the instructions, and at the same time, RNA was identified using RNA integrity by Bioanalyzer 2100 to ensure its integrity, followed by quality control of RNA samples by a NanoDrop ND-1000 spectrophotometer.

### 2.9. RT-qPCR Validation

RNA was obtained according to the above methods, with primers for tsRNA completed by the neck-loop method, followed by PCR using the Bulge-Loop TM miRNA qRT-PCR Starter Kit in a 7500 fast real-time PCR system (Applied Biosystems, Foster City, CA, USA) according to the reagent vendor’s instructions. U6B small nuclear RNA (RNU6B) was selected as an internal reference gene, and the expression of relevant tsRNA in each sample was assessed using the 2^−ΔΔCt^ method to validate the results obtained by the microarray.

### 2.10. Target Gene Prediction

We know that some tsRNAs can exert their biological functions by acting as miRNAs [20], so we tried to use the target gene prediction approach of microRNAs to help us find the target genes of tsRNA. In this study, two common prediction methods, Targetscan (www.Targetscan.org) and miRanda (www.microrna.org), were used to predict target genes. In 2003, Bartel’s team developed TargetScan, which became the first algorithm to predict vertebrate miRNA targets [21,22] and can search by gene symbols and/or species-specific miRNA names [23]. miRanda is also an old dynamic programming algorithm based on RNA secondary structure and free energy [24], but unlike most of the miRNA target predictors, miRanda considers the whole miRNA sequence matching, it considers the seed region by weighting the matches in the seed region more heavily [25]. The two algorithms have their own characteristics, so we compare them comprehensively to screen target genes.

### 2.11. Gene Ontology (GO) Annotation and Kyoto Encyclopedia of Genes and Genomes (KEGG) Pathway Enrichment Analyses

To further explore the biological function of target Genes of tsRNA associated with septic cardiomyopathy, we used Gene Ontology (GO) functional annotation and Kyoto Encyclopedia of Genes and Genomes (KEGG) pathway enrichment analyses, respectively. The GO analysis includes three subdomains: cellular component, biological process, and molecular function. We used *p* value to screen the predictors. *p* < 0.05 was used as the criterion of significant difference. Within this range, we enumerated the top 10 functional and pathway predictors.

### 2.12. Statistical Analysis

We used GraphPad Prism V8.0(GraphPad Software, CA) as statistical Software. Due to the complexity and variability of our CLP modelling and the error of the experiment, we removed outliers from the data. The data in this study were tested for normality and all conformed to a normal distribution. An F-test was also performed to test if the variances from the data of the two groups were equal. Measurement data were expressed as mean ± standard deviation and differences between the two groups were compared using an unpaired t-test, *p* < 0.05 was considered statistically significant.

## 3. Results

### 3.1. Alterations of tsRNA Expression Profiles Induced by Septic Cardiomyopathy

CLP modeling confirmed that there were varying degrees of elevation of CK-MB and LDH in the CLP Group at 12 and 24 h after modeling compared with SHAM group (Figure 1A,B). At the same time, we performed HE staining on myocardial tissue sections. The results showed that the myocardial cells in CLP group were disorganized, fragmented, and even dissolved; the myocardial texture disappeared; and the inflammatory cells infiltrated, this is consistent with the results of myocardial enzymes (Figure 1C). There was also a significant difference in mortality between the two groups. The mortality rate of CLP group mice reached 22.2% at 24 h after modeling, while SHAM group mice did not die at 24 h after modeling. All the above results indicate that CLP modeling is successful. Microarray hybridization was then used to determine the expression levels of tsRNA in the CLP group versus SHAM Group, and we designed and used a total of 1767 tsRNA probes. A total of 158 tsRNAs (*p* < 0.05, FC ≥1.5) were screened by comparing the two groups. Among them, 101 tsRNAs were up-regulated and 57 tsRNAs were down-regulated (Figure 1D,E). We then compared expression-altered tsRNAs with data collected in the tRFdb database and found that 4 of the 101 tsRNAs with up-regulated expression were included, only 1 of the 57 tsRNAs whose expression was down-regulated intersected with the tRFdb database (Figure 1G).

### 3.2. Preliminary Analysis of Microarray Results

We further analyzed the tsRNA subtypes of up- and down-regulated tsRNA. The results showed that tRF-3 (4.95%), tRF-5 (41.58%), 3′tiRNA (0.99%), 5′tiRNA (52.48%) were found in the up-regulated group; tRF-3 (14.04%), tRF-5 (54.39%), tRF-1 (1.75%), 3′tiRNA (10.53%), 5′tiRNA (19.29%) were detected in the down-regulated group. We can see that tRF-5 and 5′tiRNA account for the highest proportion of the expressed tsRNA, it is reasonable to speculate that these two types of tsRNA may play a more dominant role in the development of septic cardiomyopathy (Figure 2A–C). At the same time, we developed an interest in the tsRNA-precursors and performed a simple taxonomic statistic. We found that in the up-regulated group, tRNA-Cys-GCA, tRNA-Ala-AGC, and tRNA-Lys-CTT were the most dominant precursors, while in the down-regulated group, tRNA-Cys-GCA, tRNA-Gln-CTG, and tRNA-Pro-TGG were the most dominant. By the same comparison, we can find that tRNA-Cys-GCA is the most important precursor in up-regulated group and down-regulated group; moreover, tRNA-Cys-GCA cleavage can produce several subtypes of tsRNA including tRF-3, tRF-5, and 5′tiRNA. This suggests that tRNA-Cys-GCA is most susceptible in septic cardiomyopathy and produces multiple tsRNAs after restriction to participate in the process of septic cardiomyopathy. Why tRNA-Cys-GCA stands out from a multitude of precursors is worth our further study and thinking (Figure 2D,E).

### 3.3. Identification of tsRNA & qPCR Verification

Of the 158 tsRNAs with significant differences in expression, we further verified the microarray results, 7 up-regulated tsRNAs (5′tiRNA-34-AspGTC-2, 5′tiRNA-33-CysACA-1, 5′tiRNA-33-ProTGG-1, 5′tiRNA-32-LysCTT-11, 5′tiRNA-33-AlaTGC-7, 5′tiRNA-36-ArgTCT-1, 5′tiRNA-35-LeuCAG-4) were selected, 1 (tRF5-18-ArgCCT-3) was selected among the down-regulated, and a total of 8 tsRNAs were verified by qPCR. As shown in Figure 3, 5′tiRNA-34-AspGTC-2, 5′tiRNA-33-CysACA-1, 5′tiRNA-33-ProTGG-1, 5′tiRNA-32-LysCTT-11, 5′tiRNA-33-AlaTGC-7, 5′tiRNA-36-ArgTCT-1, 5′tiRNA-35-LeuCAG-4, and tRF5-18-ArgCCT-3 were consistent with microarray results, it was up-regulated or down-regulated after modeling (*p* < 0.05). The results of qPCR showed that the results of microarray were reliable, and the tsRNA may be involved in the development of septic cardiomyopathy, which is worthy of further analysis.

### 3.4. Targets of Septic Cardiomyopathy-Related tsRNAs

As mentioned above, tsRNA has a similar function to microRNA. So we chose to use Targetscan and miRanda for target gene prediction of the following five tsRNAs with significant differences in expression (5′tiRNA-33-AlaTGC-7, 5′tiRNA-36-ArgTCT-1, 5′tiRNA-33-CysACA-1, 5′tiRNA-34-GlnCTG-5 and 3′tiRNA-41-ProTGG-4, Figure 4A–E). The target genes of tsRNA also intersect, as shown in Figure 4F. The intersections of these target genes include Yipf6 and NCOA4. Yipf6 is a five transmembrane-spanning protein associated with Golgi compartments, and Yipf6 has been shown to be associated with spontaneous intestinal inflammation in mice [26]. NCOA4 (nuclear receptor coactivator 4) mediates the selective autophagic degradation of the cytosolic iron storage complex ferritin [27], which has been shown to be associated with mitochondrial DNA damage [28], mitochondrial DNA damage is also common in septic cardiomyopathy; therefore, NCOA41 may play a potential role in septic cardiomyopathy.

### 3.5. Potential function of Septic Cardiomyopathy-Related tsRNAs Revealed by GO and KEGG Analysis

To further investigate the potential biological function of septic cardiomyopathy-related tsRNAs, we selected to perform Gene Ontology (GO) annotation and Kyoto Encyclopedia of Genes and Genomes (KEGG) pathway enrichment analyses. The three subdomains of GO analysis include cellular component, biological process, and molecular function, and the top ten enriched terms for tsRNA associated with septic cardiomyopathy in the three subdomains after GO analysis are shown in Figure 5A–C. In terms of cellular component, tsRNAs are mainly present in cellular anatomical entities, and are mainly involved in cellular processes in biological processes. Finally, protein binding is significantly enriched in molecular functions. The results of the KEGG pathway analysis suggested that septic cardiomyopathy-related tsRNAs may play a key role in the development of septic cardiomyopathy through the Wnt signaling pathway and MAPK signaling pathway (Figure 5D). The Wnt signaling pathway can activate non-canonical pathways through calcium/calmodulin-dependent kinase II (CAMKII), it leads to sustained up-regulation of inflammatory cytokines, including IL-12, IL-6, IL-8, IL-1β, and macrophage inflammatory protein-1β (mip-1β) activation, involved in the development of septic cardiomyopathy [29]. As for the MAPK pathway, it has been shown that sepsis-associated inflammatory factors can promote endothelial cell activation, dysfunction, and apoptosis through activating the p38 MAPK pathway regulated by NF-ΚB signal, and thus participate in sepsis [30].

### 3.6. Function Detection of Septic Cardiomyopathy-Related tsRNAs after ANG Interference

Since the Gene Ontology enrichment analysis mentioned above shows that the biological processes in which tsRNAs are involved are mainly cellular processes. The cellular process involves multiple stages and aspects of the cell and is relatively broad in scope, so we chose to use CCK-8 as well as LDH (cell death) as the detection methods after interfering with the key enzyme of tsRNA production, angiogenin (ANG), to observe the effect of LPS combined with TNF-α on H9C2 cells. We first confirmed whether the si-ANG was valid (Figure 6A). After si-ANG screening, we considered whether the expression of previously validated tsRNA would change after ANG interference, and we randomly selected 5′-tiRNA-33-CysACA-1 for qPCR again, with the results as expected; 5′-tiRNA-33-CysACA-1 expression decreased after ANG was perturbed and was statistically significant (Figure 6B). We then further examined the effect on cells after ANG interference, as shown in Figure 6C,D. After 24 h of combined LPS Plus TNF-α treatment, the cell activity was significantly decreased in the ANG interference group compared with the NC Group, and the release of LDH (cell death) increased obviously. The results suggest that tsRNA may be involved as a protective factor during the combined treatment of LPS and TNF-α. Therefore, we speculate that tsRNA may play a protective role in the biological process of the development of septic cardiomyopathy, but the specific mechanism, we need to further explore.

## 4. Discussion

Sepsis is a common, fatal, and expensive disease worldwide. Although sepsis has long been recognized, it was not clinically defined until the late 20th century because of the lack of effective antimicrobial agents and supportive care, patients with sepsis cannot survive long enough or have sequelae of organ dysfunction [31]. Sepsis-induced myocardial dysfunction (SIMD) is a disease characterized by the intrinsic systolic and diastolic dysfunction of the myocardium during sepsis, with a poor clinical prognosis and high mortality. Many pathophysiological mechanisms, such as mitochondrial dysfunction, abnormal immune response, metabolic reprogramming, excessive production of reactive oxygen species (ROS), calcium regulation disorder, etc., are involved in the development of SIMD [32]. In order to study the complex molecular mechanism of sepsis and myocardial dysfunction, many animal models of sepsis have been established. The mouse model of cecal ligation and puncture (CLP) has been recognized as the most commonly used animal model because of its similarity to the development and characteristics of human sepsis. In this model, sepsis originates from multiple microbial infections within the abdominal cavity, followed by bacterial translocation into the blood cavity, further triggering a systemic inflammatory response and involving multiple organs throughout the body [33]. Based on the above understanding, we used CLP mouse model for microarray analysis to gain a more comprehensive and extensive understanding of sepsis and septic cardiomyopathy.

In 2008, tsRNA was discovered and named by researchers in Giardia Lamblia [8]. In 2009, researchers reported the first functional tsRNA, named tRF-1001, and argued that tsRNA is not a random byproduct of tRNA degradation or biogenesis, but rather an abundant and novel small RNA. It has precise sequence structure, specific expression pattern, and specific biological function [34]. Since then, the related functions of tsRNA have been widely studied in various biological diseases, and become another research hotspot in small non-coding RNAs. To date, the clinical value of tsRNA has been discussed in a variety of tumors, and the development of some cardiovascular diseases has also been shown to be associated with certain tsRNA. For example, tRFs are involved in isoprenaline myocardial hypertrophy in Sprague-Dawley rats, perhaps specifically because tRF-5 binds to the 3′ UTR of the hypertrophy regulator TIMP3 mRNA and inhibits its expression, which leads to hypertrophy of cardiac myocytes [35]. In addition, Yang et al. screened 77 tsRNAs that might be involved in rheumatic heart disease with atrial fibrillation by RNA sequencing, providing a reference for us to explore the regulatory role of tsRNA in rheumatic heart disease with atrial fibrillation [36]. tsRNA has also been studied in fulminant myocarditis [37], myocardial ischemia [38], and arteriosclerosis [39]. Interestingly, however, the association of tsRNA and septic cardiomyopathy has not been studied. Our previous work has explored the association between microRNA and doxorubicin-induced cardiac toxicity [40]; therefore, we have been looking at non-coding RNA such as tsRNA mentioned in this article.

In this study, we first used CLP model mice and screened 158 tsRNAs with significant changes in expression by microarray hybridization, of which 101 had elevated expression and 57 had decreased expression. We have classified the tsRNA whose expression has changed and found tRF-5 and 5′ tiRNA subclasses were the most abundant. At the same time, we also performed some simple statistics on tsRNA precursor, and found that tRNA-Cys-GCA was the most important tsRNA precursor. We, therefore, hypothesize that tRNA-Cys-GCA is susceptible in the situation of septic cardiomyopathy and is cleaved in this stressful environment. This is consistent with the findings of Green et al. [41] in tsRNA-related studies in osteoarthritis and Fu et al. [42] in tsRNA-related studies in aortic coarctation. In their study tRNA-Cys-GCA was also an important precursor for the production of tsRNA. tRNA-Cys-GCA seems to be active in inflammatory or heart-related diseases and deserves further investigation. Furthermore, tRF-5 and 5′-tiRNA may play an important role in the development of septic cardiomyopathy. Then we selected eight tsRNAs with the most obvious expression changes for qPCR validation of microarray results, and all of them were consistent with microarray results. This demonstrated the reliability of the microarray results and suggested that these eight tsRNAs may have some biological function in septic cardiomyopathy. Next, we selected five tsRNAs with the most significant differences in expression, predicted their target genes using Targetscan and miRanda, and explored target gene function using GO analysis and KEGG analysis. The results of GO analysis indicated cellular process, while the analysis of KEGG pathway suggested that the Wnt pathway and the MAPK pathway might be involved in the development of septic cardiomyopathy. We know that the Wnt pathway is involved in septic cardiomyopathy by regulating mitochondrial mPTP and calcium ion load [43], and the MAPK pathway may act on septic cardiomyopathy by affecting mitochondrial damage and endoplasmic reticulum (ER) dysfunction [44]. Therefore, tsRNA may be involved in the regulation of septic cardiomyopathy through these related pathways. Finally, we consider that in the biological process of GO analysis, the concept of cellular process in which tsRNA is involved is broad, so we further demonstrated the role of tsRNA in septic cardiomyopathy by interfering with Angiogenin(ANG), the upstream key enzyme that generates tsRNA, and performing CCK-8 detection with LDH (cell death) detection. We chose ANG for interference because a large number of studies have demonstrated that ANG cleaves the conserved 3′-CCA end of tRNA or the tRNA anticodon loop to form tiRNA in response to various stresses, such as hunger stress and oxidative stress [13,14,45,46], and ANG-mediated tiRNA production is precisely regulated. For example, tiRNA production is influenced by tRNA modification and stress intensity [47,48]. The expression of tiRNA changed significantly in our microarray results, and we speculated that it might be closely related to septic cardiomyopathy, so we chose to treat its upstream nuclease ANG. The results were as expected. The production of tiRNA was decreased after the interference of ANG. At the same time, the activity of cells was decreased and the release of LDH was increased after the combination of LPS and TNF-α, thus, it is confirmed that tsRNA, especially tiRNA, plays a protective role in septic cardiomyopathy.

Of course, this study also has the following limitations. First, the sample size is small, and further studies of the mechanism need to be supported by larger samples. Second, although CLP is a well-established animal model of sepsis, there are differences relative to clinical samples and clinical samples need to be collected to validate the statement. Finally, in the function aspect, it is mainly through the biological information method to predict, but still lacks the further experiment.

## 5. Conclusions

To sum up, we revealed the change of tsRNA expression level after CLP modeling by microarray, which suggests that tsRNA may be a key factor in the development of sepsis and septic cardiomyopathy and may play a role in protecting myocardial injury. Our study has opened up a new perspective for the follow-up study of sepsis and septic cardiomyopathy molecular mechanism of reference.

## Figures and Tables

**Figure 1 genes-13-02258-f001:**
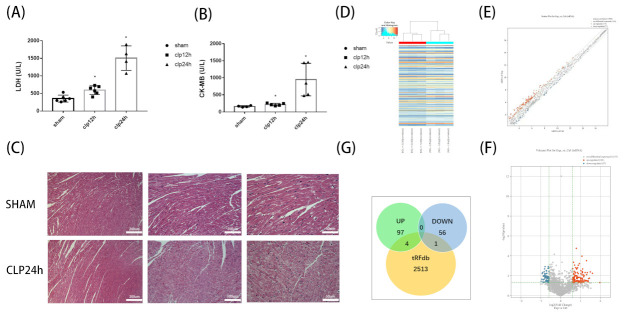
Alterations of tsRNA expression profiles induced by septic cardiomyopathy. (**A**) Comparison of CK-MB in myocardial tissue of CLP group versus SHAM group. * *p* < 0.05. (**B**) Comparison of LDH in myocardial tissue of CLP group versus SHAM group. * *p* < 0.05. (**C**) Comparison of myocardial HE staining between CLP Group and SHAM group. (**D**) Septic cardiomyopathy-related tsRNAs were shown in the heat map. The colors in the panel indicate the relative expression levels (log2 transformation). The color bar graph on the top panel shows the sample group at the top. (**E**) The scatter plot showed that there were 158 significantly altered tsRNAs between CLP and SHAM groups (*p* < 0.05, FC ≥ 1.5). (**F**) The volcano plot showed that there were 158 significantly altered tSNAs between CLP and SHAM groups (*p* < 0.05, FC ≥ 1.5). Red showed up-regulated tsRNAs and blue showed down-regulated tsRNAs. Data were presented as the mean ± standard deviation (*n* = 3). (**G**) Venn diagram indicates the relationship between up- and down-regulated expression groups and tsRNAs that have been studied in tRFdb. CLP: Cecal ligation and puncture; tRF: tRNA fragments; tiRNA: tRNA halves; tsRNA: tRNA-derived small RNA.

**Figure 2 genes-13-02258-f002:**
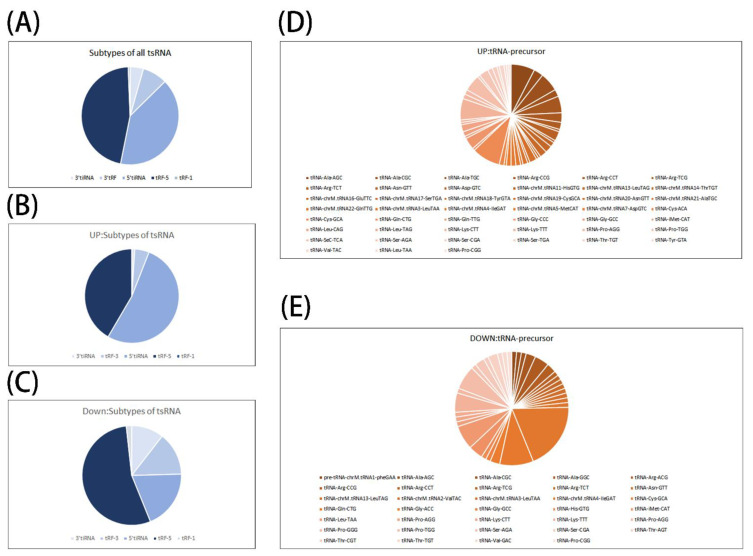
Preliminary analysis of microarray results. (**A**–**C**) Pie chart of the distribution of tiRNA and tRF subtypes. The color represents the tiRNA and tRF subtypes. (**D**,**E**) Pie chart of the distribution of tsRNA-precursors. The color represents the tsRNA-precursors. tRF: tRNA fragments; tiRNA: tRNA halves; tsRNA: tRNA-derived small RNA.

**Figure 3 genes-13-02258-f003:**
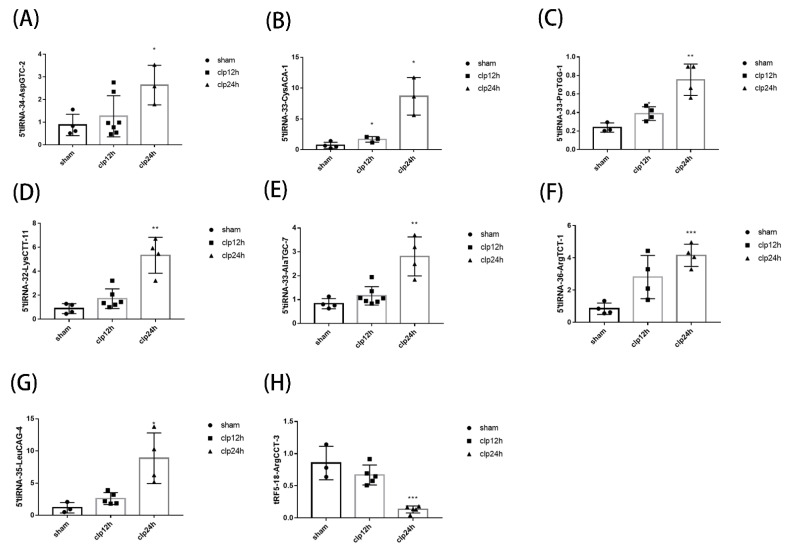
Identification of septic cardiomyopathy-related tsRNAs and qPCR verification. (**A**–**H**) The qPCR results confirmed that 5′tiRNA-34-AspGTC-2 (*n* = 4 in SHAM group, *n* = 7 in CLP 12 h group, *n* = 3 in CLP 24 h group), 5′tiRNA-33-CysACA-1 (*n* = 4 in SHAM group, *n* = 3 in CLP 12 h group, *n* = 3 in CLP 24 h group), 5′tiRNA-33-ProTGG-1 (*n* = 3 in SHAM group, *n* = 4 in CLP 12 h group, *n* = 4 in CLP 24 h group), 5′tiRNA-32-LysCTT-11 (*n* = 4 in SHAM group, *n* = 6 in CLP 12 h group, *n* = 4 in CLP 24 h group), 5′tiRNA-33-AlaTGC-7 (*n* = 4 in SHAM group, *n* = 7 in CLP 12 h group, *n* = 4 in CLP 24 h group), 5′tiRNA-36-ArgTCT-1 (*n* = 4 in SHAM group, *n* = 4 in CLP 12 h group, *n* = 4 in CLP 24 h group), 5′tiRNA-35-LeuCAG-4 (*n* = 3 in SHAM group, *n* = 5 in CLP 12 h group, *n* = 4 in CLP 24 h group) and tRF5-18-ArgCCT-3 (*n* = 4 in SHAM group, *n* = 5 in CLP 12 h group, *n* = 5 in CLP 24 h group) were consistent with tsRNA microarray data. Data were presented as mean ± standard deviation. * *p* < 0.05; ** *p* < 0.01; *** *p* < 0.001 compared with the SHAM group. CLP: Cecal ligation and puncture; qPCR: Quantitative real-time PCR; tRF: tRNA fragments; tiRNA: tRNA halves; tsRNA: tRNA-derived small RNA.

**Figure 4 genes-13-02258-f004:**
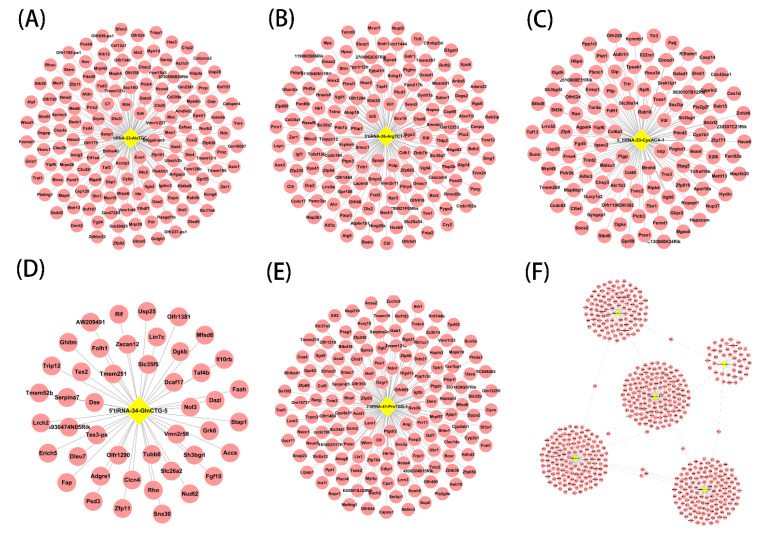
Targets of septic cardiomyopathy-related tsRNAs. (**A**–**F**) The targets of 5′tiRNA-33-AlaTGC-7, 5′tiRNA-36-ArgTCT-1, 5′tiRNA-33-CysACA-1, 5′tiRNA-34-GlnCTG-5 and 3′tiRNA-41-ProTGG-4 are shown. tiRNA: tRNA halves; tsRNA: tRNA-derived small RNA.

**Figure 5 genes-13-02258-f005:**
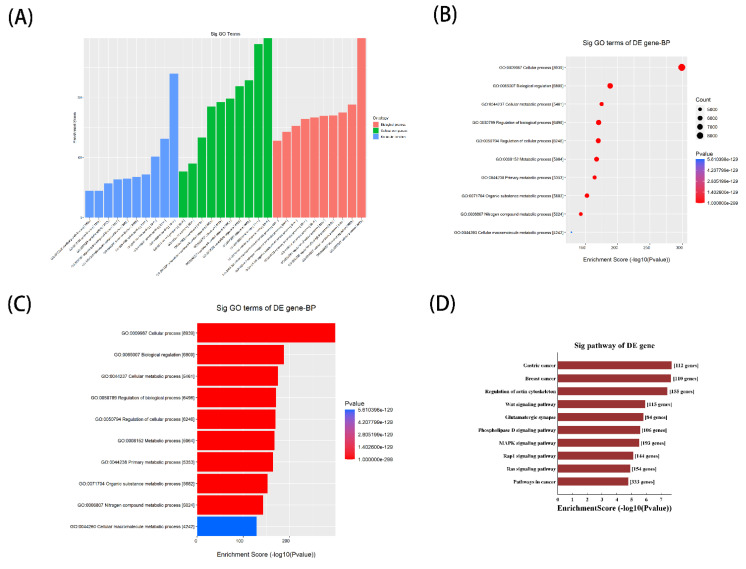
Potential function of septic cardiomyopathy-related tsRNAs revealed by GO and KEGG analysis. (**A**)–(**D**) Gene Ontology enrichment analysis and Kyoto Encyclopedia of Genes and Genomes pathway analysis for septic cardiomyopathy-related tsRNAs. BP: biological process; CC: cellular component; DE: differently expressed; GO: Gene Ontology; MF: molecular function; KEGG: Kyoto Encyclopedia of Genes and Genomes.

**Figure 6 genes-13-02258-f006:**
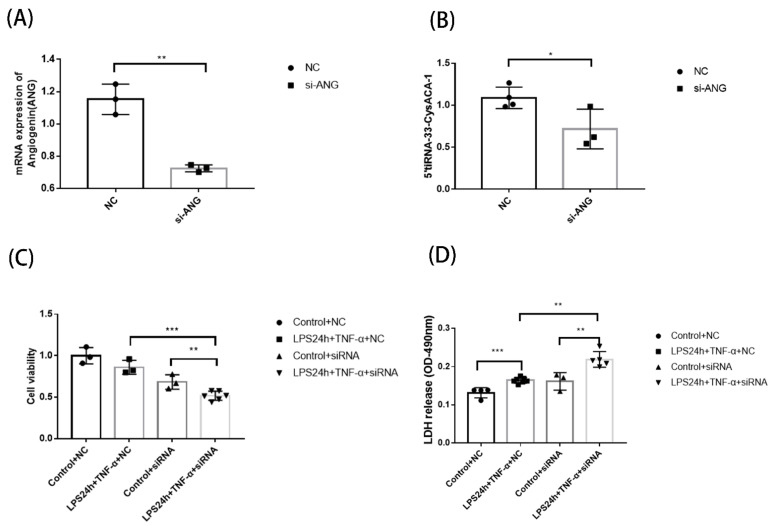
Function detection of septic cardiomyopathy-related tsRNAs after ANG interference. (**A**) The effect of ANG interference is detected by qPCR (*n* = 3 in NC group, *n* = 3 in si-ANG group) (**B**) The qPCR results confirmed that the expression of 5′tiRNA-33-CysACA-1 (*n* = 4 in NC group, *n* = 3 in si-ANG group) changed after ANG was interfered. Data were presented as mean ± WB standard deviation. * *p* < 0.05 compared with the NC group. (**C**) The effect of si-ANG on cell viability after 24 h of LPS + TNF-α treatment ** *p* < 0.01; *** *p* < 0.001 compared with the Control group. (**D**) The effect of si-ANG on the release of LDH (cell death) after 24 h of LPS + TNF-α treatment ** *p* < 0.01; *** *p* < 0.001 compared with the Control group. si-ANG: ANG-specific siRNA; NC: negative control; qPCR: Quantitative real-time PCR; tRF: tRNA fragments; tiRNA: tRNA halves; tsRNA: tRNA-derived small RNA.

## Data Availability

The datasets presented in this study can be found in online repositories. The names of the repository/repositories and accession number(s) can be found below: Gene Expression Omnibus (GEO) database under accession number GSE214251. (https://www.ncbi.nlm.nih.gov/geo/query/acc.cgi?acc=GSE214251).

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
