# Peer review of "Microarray Analysis Reveals Changes in tRNA-Derived Small RNAs (tsRNAs) Expression in Mice with Septic Cardiomyopathy"

_genes, 2022, doi:10.3390/genes13122258_

Round 1
Reviewer 1 Report
This study performed in C57BL/6 mice analyzed tRNA-derived small RNAs (tsRNAs) expression in mice with septic cardiomyopathy. The data are very well described. Certain issues should be addressed?1. The analysis has been performed lege artis. Nevertheless, a very fundamental question arises. Are the observed changes causally related to the cardiac dysfunction in septic cardiomyopathy or do they reflect to a large extent epiphenomena? An epiphenomenon is a consequence of the primary phenomenon but cannot affect the primary phenomenon.
2. Statistical analysis. Outliers were removed. How do you define an outlier? It is very tricky to remove outliers. 3. Statistical analysis. How do you define an independent experiment? Is this one mouse? This seems impossible in relation to the microarray analysis. 4. There are several substains of C57BL/6. Were these C57BL/6J, C57BL/6N from Taconic, or C57BL/6N from Charles River? 5. Two abbreviations are unexplained in the abstract: Gene Ontology (GO) annotation and Kyoto Encyclopaedia of Genes and Genomes (KEGG) pathway enrichment analyses. I know this may be difficult but many do not know these abbreviations.
Reviewer 2 Report
In the manuscript submitted by Ludong Yuan et al and colleagues, the authors have attempted to reveal the changes in tRNA-derived small RNAs (tsRNAs) expression in mice with septic cardiomyopathy. The manuscript provided the modelistic approach to show the differential expression of tsRNAs by microarray analysis, validated by qPCR followed by GO and KEGG analysis of the target genes statistically showing the protective role of tsRNAs in septic cardiomyopathy. The manuscript consists of bioinformatics and wet lab experiments inclination. Overall experimental planning, data analysis is fine.
There are a few major concerns that need to be addressed. My comments are as follows.
1. The author claimed that tRNA-Cys-GCA is the most important precursor in up-regulated and down-regulated groups. This is not validated by any wet lab experiment showing the common expression of tRNA-Cys-GCA precursor
2. As tRNA-Cys-GCA was found to be the most important precursor in the up- and the downregulated tRNA fragments, its target gene prediction and further GO and KEGG pathway analysis should also be carried out in order to further confirm its importance.
3. As mentioned, tRNA-Cys-GCA generates 5’tRF and 5’tiRNA in abundance. These tsRNA generation should also be validated with ANG interference experiments for the functional validation.
4. The images in the manuscript are so miniaturised therefore the dpi of the images needs to be increased to get the more resolution.
Round 2
Reviewer 2 Report
The manuscript can be accepted for publication.